# How Downward and Upward Comparisons on Facebook Influence Grandiose and Vulnerable Narcissists’ Self-Esteem—A Priming Study

**DOI:** 10.3390/bs11030039

**Published:** 2021-03-19

**Authors:** Phillip Ozimek, Hans-Werner Bierhoff, Elke Rohmann

**Affiliations:** 1Department of Psychology, University of Hagen, Universitaetsstr. 27, 58084 Hagen, Germany; 2Department of Psychology, Ruhr University Bochum, Universitaetsstr. 150, 44801 Bochum, Germany; hans.bierhoff@rub.de (H.-W.B.); elke.rohmann@rub.de (E.R.)

**Keywords:** grandiose narcissism, vulnerable narcissism, social comparisons, priming, Facebook

## Abstract

Past research showed that social networking sites represent perfect platforms to satisfy narcissistic needs. The present study aimed to investigate how grandiose (GN) and vulnerable narcissism (VN) as well as social comparisons are associated with Facebook activity, which was measured with a self-report on three activity dimensions: Acting, Impressing, and Watching. In addition, the state self-esteem (SSE) was measured with respect to performance, social behavior, and appearance. One hundred and ten participants completed an online survey containing measures of SSE and Facebook activity and a priming procedure with three experimental conditions embedded in a social media context (upward comparison, downward comparison, and control group). Results indicated, as expected, that high VN was negatively associated with SSE on each subscale and the overall score. In addition, it was found that VN, but not GN, displayed positive associations with frequency of Facebook activities. Finally, it was proposed and confirmed that VN in interaction with the priming of downward comparisons negatively affected SSE. The conclusion drawn is that VN represents a key variable for the prediction of self-esteem as well as for the frequency of Facebook activity.

## 1. Introduction

In the following, the constructs Facebook activity, social comparison, and narcissism are described.

### 1.1. Facebook Activity

Facebook is the most popular social networking site (SNS) in the world: It counted more than 2.6 billion monthly active users in the first quarter of 2020—a number even higher than the one reported in the year before [1].

Research has recently identified potential motives behind Facebook usage. Most of these motives fall into the categories of social exchange, online identity construction and self-presentation, and watching other people [2,3,4,5]. Or, as Nadkarni and Hoffman [6] pointed out in their dual-factor model of Facebook use, the utilisation of this SNS is “primarily motivated by two basic social needs: (1) *the need to belong*, and (2) the *need for self*-*presentation”* (p. 3). These factors may be supplemented by adding a third need, namely, that of *social comparison* [7,8]. Which of these motives accounts for users’ inclination to use Facebook, however, seems to highly depend on their personality traits: For instance, extraversion, neuroticism, depression, loneliness, sensation seeking, and narcissism have been found to be positively related to SNS activity [9,10,11]; for a review, see [12].

Ozimek, Baer, and Förster [13] integrated these findings into a comprehensive framework: According to their Social Online Self-Regulation Theory (SOS-T), the different motivations to use Facebook all fall within the broader term of self-regulation as the primary goal. More specifically, Facebook seems to serve as a means to reach multiple self-regulatory goals, such as increasing self-esteem, presenting oneself, or interacting with others. Even though more work is needed in this respect, empirical evidence supports SOS-T: Several studies [14,15] demonstrated that both for users high in materialism, i.e., a personality facet triggering the idea that acquiring possession leads to happiness [16], and for those high in VN, Facebook seems to serve as a means to self-regulation.

### 1.2. Social Comparisons and Facebook

However, several issues in this field of research still remain unresolved, among them the question of whether social comparisons may prompt users’ engagement in Facebook activity in order to self-regulate. More specifically, the distinction between upward and downward comparison should be investigated with respect to their impact on self-esteem. Whereas an upward comparison is directed toward a person who is performing better than oneself, a downward comparison is directed toward a person who is performing worse than oneself. In the current study, the influence of downward and upward comparisons in the context of the SNS Facebook was experimentally investigated. More specifically, situational manipulations of the availability of upward and downward comparisons, respectively, were employed in order to measure their impact on the self by focusing on state self-esteem. The availability of upward and downward comparisons marks the upper and lower boundary of a comparison standard. Therefore, their systematic manipulation includes the upper and lower boundary of feedback on self-esteem via social comparison processes. Meta-analytic results indicate that both upward and downward comparisons exert significant influences on self-esteem [17].

Meanwhile, social comparison behavior in the offline world constitutes a well-examined phenomenon: It was in 1954 that Leon Festinger published his *A Theory of Social Comparison Processes*. According to that theory and its subsequent elaborations [17,18,19,20,21] such social comparisons can happen either in a conscious or automatic way; they may refer to a variety of categories; and they can involve any person an individual is confronted with. The choice of the comparison person depends not least on the motivation for the comparison process [21]. For example, if the person is interested in improving his or her competence in algebra, the choice of a comparison person who successfully solves algebraic tasks is obvious.

Importantly, Festinger [22] draws a distinction between upward, downward, and lateral comparisons (see also [17]). Such comparisons are usually focused on subjectively important categories (e.g., reaching high achievements, doing sports activities). Participants are asked to compare themselves with another person whom they consider as equal (lateral comparison), superior (upward comparison), or inferior (downward comparison) on these categories. Within Facebook, the profiles of other members provide information about such categories of life and the standing of the other members, eliciting spontaneous comparisons of the users referring to their own standing (above or below the standing of other members) on the categories considered.

It seems that downward comparisons tend to positively influence self-evaluation and affect, whereas upward comparisons tend to threaten self-evaluation and prompt negative affect, even though this rule of thumb does not apply to every individual and every situation [23,24,25,26,27,28]. For example, students who feared that they would be confronted with the same academic difficulties as the target person they had read about assessed their academic success as lower than participants in control conditions. Therefore, their self-confidence was impaired “when they drew parallels between themselves and the target” [29] (p. 350). As a consequence, for these participants who felt that they were vulnerable because they were threatened by the scenario description of academic failure downward comparisons backfired.

In the Facebook context, attractive profiles of others caused stronger user feelings of inferiority than unattractive profiles [30].

Another branch of research investigated interindividual differences in social comparisons. Specifically, Gibbons and Buunk [31] investigated the construct of Social Comparison Orientation (SCO)—a variable describing how much an individual is inclined to compare with others. Evidence speaks to a two-dimensional structure of SCO: A distinction can be drawn between people’s inclination to either compare their abilities and skills (ability-based SCO) or their opinions and attitudes (opinion-based SCO) to those of others [31,32].

As the need to compare has also been proved to be an important motivation for higher Facebook activity [6,7,8], more recent studies have investigated social comparisons in the online world. In this respect, Lee [7] found that users displaying a high SCO also tend to compare themselves more often on Facebook, and that the frequency of social comparisons on Facebook is positively associated with the experience of negative affect. His investigation was, however, limited to correlations.

Overcoming that limitation, several studies [33,34] succeeded in demonstrating a direct negative effect of upward comparisons, as elicited by exposure to SNS profiles, on self-evaluation and state of self-esteem (SSE). In addition, Valkenburg et al. [35] demonstrated that positive versus negative feedback on the profiles of SNS users increased versus decreased both their self-esteem and their well-being. However, the method of providing users with direct profile-feedback seems to hardly reflect social comparison activities that naturally occur in SNS environments: The latter, in turn, consist, rather, of watching other users [8].

An investigation of self-esteem with regard to social comparison processes in the SNS context, by means of a paradigm that guarantees a high relevance of the comparison process to each participant, has (to the authors’ knowledge) not been conducted yet. Our study tries to fill that gap by establishing either upward or downward social comparisons with a person known by, and in a category that is subjectively important to, the participant. As mentioned above, individual differences in personality traits may alter the way people react on social comparisons. In this respect, the personality trait of narcissism seems to be particularly promising.

### 1.3. Narcissism and Its Associations with Social Comparisons and Facebook Activity

The personality trait of narcissism is characterized by an inflated self-view, entitlement, and increased egocentrism [36,37,38]. Wink [39] identified two dimensions of the narcissistic personality, namely, grandiose (GN) and vulnerable narcissism (VN). According to Wink’s work [39] and several follow-up studies [40,41,42,43,44], GN and VN share a common core but also differ in several characteristics: GN, on the one hand, is associated with an exhibitionistic tendency, extraversion, dominance, arrogance, a high approach orientation, impulsivity, and the insistence on one’s own needs. This personality trait is described in detail by Campbell in his (extended) agency model [45,46]. Of special relevance for this study are (a) grandiose narcissists’ selfish, success-oriented self-view in agentic (but not communal) domains; (b) their typically narcissistic interpersonal skills (e.g., charisma, extraversion); and (c) the use of intrapsychic and interpersonal strategies (such as self-enhancement) in order to maintain their inflated self-view (i.e., as a means of self-regulation) [47,48]. Concerning the relationship between GN and frequency of social comparisons, it has been demonstrated that GN is positively correlated with the frequency of—especially downward—comparisons and subsequent experience of increased positive affect, as demonstrated by Krizan and Bushman [49], which is in line with Campbell and Foster’s [46] extended agency model.

VN, on the other hand, is characterised by “defensiveness, hostility, sensitivity to slight, and concern with one’s own adequacy” and by “introversion, discomfort in leadership roles, and lack of self-confidence in social settings” [39] (p. 596). Furthermore, VN is marked by an interdependent and unstable self-view, concerns about being liked by others, an increased anxiety, a high avoidance orientation, and a tendency toward negative affectivity [40,41,43,50]. Unlike their grandiose counterparts, vulnerable narcissists do not regularly use self-enhancement strategies but depend heavily on feedback from their social environment to regulate their self-esteem—which is, however, rather difficult to obtain for these individuals due to their tendency towards hostility and anxiety in social relationship leading to social withdrawal from others and due to their experience of interpersonal conflicts [40,41,42,44]. In addition, Ozimek et al. [51] among others reported positive correlations of VN with SNS activity. The present study aimed to confirm and extend these results.

Lockwood [29] demonstrated that high vulnerability led to reduced self-evaluation as a consequence of downward comparison. Although vulnerable narcissists might have inferred superiority from being better than the target persons, they were irritated by the perceived threat of their own failure. Such a threat looms behind downward comparisons in the context of an anxious expectation of possible failure. Perceived vulnerability tends to be high in high scorers on VN. Therefore, they are likely to routinely assume that the bad fate of similar others might happen to themselves. As a consequence, their self-confidence is likely to be impaired. The results of Besser and Priel [52] agree with this conclusion by relating individual differences in vulnerable narcissism to downward comparisons.

## 2. Materials and Methods

### 2.1. Hypotheses

In correspondence with previous research [42], it was assumed that GN and VN show opposite associations with self-esteem.

**Hypothesis** **1.**
*Grandiose narcissism is positively correlated with SSE (after statistically controlling for vulnerable narcissism, i.e., partialling out VN) whereas vulnerable narcissism is negatively related to SSE (after statistically controlling for grandiose narcissism; i.e., partialling out GN).*


The second hypothesis also focuses on SSE taking social comparisons—more specifically downward comparisons—into account. It was inspired by research on the consequences of vulnerability on self-ratings after downward comparisons [29]. Specifically, the following hypothesis is investigated:

**Hypothesis** **2.**
*The influence of downward social comparisons on participants’ SSE is moderated by VN. The higher the VN of respondents, the lower their situational self-esteem after the elicitation of downward comparisons.*


To measure the self-relevant effects of threat of interpersonal rejection participants’ SSE was measured. SSE was focused on which should be more malleable by situational manipulations than dispositional self-esteem. SSE is likely to tap situational influences which are the result of experimental manipulations whereas dispositional self-esteem is likely to be less sensitive to situational manipulations.

Next, the prediction of SNS activity by narcissism follows.

**Hypothesis** **3.**
*VN displays a significant positive association with Facebook activity, even after control for participants’ GN.*


**Hypothesis** **4.**
*GN displays a significant positive association with Facebook activity after partialling out participants’ VN.*


**Hypothesis** **5.**
*The postulated association between VN and Facebook activity is stronger than the association between GN and Facebook activity if both correlations are controlled for each other.*


The present study also aimed to build a bridge between research on the linkages of narcissism with Facebook activity on the one hand and on associations of social comparisons with Facebook activity on the other hand. In this respect, it is interesting to investigate the roles of GN and VN in influencing the effects of upward social comparisons on Facebook activity.

Thus, in the context of upward social comparisons, grandiose, but not vulnerable, narcissists should be more inclined to use Facebook as means for self-regulation and to employ self-protective strategies. In order to investigate this issue, the following research question (RQ) was asked:

RQ 1: In how far is the influence of upward social comparisons on Facebook activity moderated by the extent of an individual’s (a) VN and (b) GN?

### 2.2. Procedure

In order to investigate the hypotheses and the RQ, an online survey was created which was answered by adult Facebook users living in Germany.

First, an online survey was created, which was available on the platform Unipark. Participants were recruited via flyers distributed at the Ruhr University Bochum as well as at universities in neighboring cities (i.e., Dortmund, Hagen, Cologne) via Facebook posts and personal addresses. Because an experimental design was employed, the internal validity of the study is quite high as participants were allocated to conditions by chance. With respect to external validity, the online survey is likely to foster the generalizability of the results because participants were recruited from a large heterogeneous data base.

The survey consisted of (1) a cover letter, including information on anonymity and voluntariness of participation, as well as a cover story about the aims of the study; (2) questions about demographic data and Facebook usage; (3) a priming procedure; (4) scales assessing the psychological constructs under investigation; and (5) four debriefing questions.

#### 2.2.1. Demographic Characteristics and SNS Usage

To assess demographic characteristics, a set of items was integrated into the survey, including questions about age, gender, occupation, highest educational degree, and relationship status. Questions on SNS usage included whether each participant owned a Facebook account, another SNS account (yes or no), and whether he or she used Facebook for private or commercial purposes or both.

#### 2.2.2. Priming Conditions

Priming refers to the fact that the way how people feel and think with respect to a certain category is determined by the accessibility of information relevant to that category. Hence, rendering either upward or downward social comparison information more salient would increase the accessibility of feelings of inferiority versus superiority (for a review, see Higgins, 1989). Participants were randomly assigned to one of three conditions: upward comparison (experimental group 1; *n* = 33), downward comparison (experimental group 2; *n* = 34), and control group (*n* = 43) (Note that the unevenness of group sizes was accounted for by the exclusion of seven participants from the analyses, which resulted in more drop-outs in the two experimental groups than in the control group). The priming procedure conducted with the two experimental groups consisted of a priming task, two cover questions, and a recall task at the end of the survey. More specifically, in experimental groups 1 and 2, upward and downward social comparisons in a subjectively important category were primed, respectively. To do that, first, participants were asked to arrange eight given categories according to subjective importance in their individual lives (i.e., “Please order the following categories with respect to the question, how important they are in your own life?”). These categories were having a satisfactory relationship; being financially successful; reaching high achievements; having healthy nutrition habits; doing sports activities; being physically attractive; being socially popular; and engaging in volunteer work/being politically active. In the next step, participants were told that one of the first three categories in their personal ranking order would be randomly chosen, about which they would have to answer some questions throughout the whole survey. Then, two cover questions about the subjectively most important category were asked (“Since how many years have you been preoccupied with category 1?”; “In which area of life are you confronted with the category most often?”). Next, the priming itself was initiated by asking participants (1) to remember five persons whom they perceived as superior (upward condition) or inferior (downward condition) in the category of life that was subjectively most important to them, including the relationship they had to these persons (e.g., “brother”, “best friend”); and (2) to recall some feelings and thoughts associated with such a moment of inferiority (upward condition) or superiority (downward condition). In the final part of the questionnaire, another cover question was asked (“Do you still know which category you placed second in your ranking order?”). The participants in the control group did not receive any priming. They only answered the remaining part of the survey.

#### 2.2.3. Inventory Measures

*Grandiose Narcissism.* In order to assess GN, the German Narcissistic Personality Inventory-15 (NPI-15) [53] was used. It constitutes of a short version of the Narcissistic Personality Inventory (NPI) [38] consisting of 15 forced-choice items with two answer options. Sample statements include: “I am more capable than other people.” (narcissistic) and “There is a lot that I can learn from other people.” (non-narcissistic). The NPI-15 turned out to have a satisfying internal consistency (Cronbach’s α between 0.73 and 0.82) and is highly correlated (*r* = 0.90) with the longer version of the NPI [53] In the current study, the internal consistency of the scale is satisfactory.

*Vulnerable Narcissism*. To assess VN, the Narcissistic-Inventory Revised (NI-R) [54] was used. It is an adapted German version of the Narcissistic-Inventory (NI) [55] and consists of 42 items that must be answered on a 5-point Likert scale, with answer options ranging from “not true at all“ to “entirely true“ (e.g., “Other people would be really amazed if they knew about my talents.”). Neumann and Bierhoff [54] reported a high internal consistency (Cronbach’s α = 0.93). In the current study the internal consistency of the scale is good.

*Facebook Activity Questionnaire*. The FAQ [8,56], represents a behavioral report of Facebook use which makes a threefold distinction between *Watching* (e.g., “I’m looking at other’s relationship status”), *Impressing* (“I’m struggling to decide which profile picture I would like to post”), and *Acting* (“I’m posting photographs”). The FAQ consists of 30 items that must be answered on a 5-point Likert scale ranging from “never” (1) to “very often” (5). The three subscales were derived from a dimensional analysis of the FAQ. Whereas *Watching* focuses on passive Facebook use, *Impressing* and *Acting* represent active modes of Facebook use. The reliabilities of the FAQ subscales were quite satisfying, both in the previous study (Cronbach’s α_Watching_ = 0.83; α_Impressing_ = 0.79; α_Acting_ = 0.77) and in the current study.

*State Self-Esteem*. SSE was assessed using a shortened German version of the State Self-Esteem Scale (SSE Scale) [57]. It consists of 15 items that have to be answered on a 5-point Likert scale, with options ranging from 1 = “not true at all” to 5 = “very much true”. It can be divided into three subscales (*Performance*, *Social*, and *Appearance*). The authors reported satisfactory reliabilities for each subscale (SSE *Performance:* Cronbach’s α = 0.80; *Social* and *Appearance:* α = 0.83, respectively). In the current study, the reliabilities were satisfactory, although the internal consistency of SSE_Per_ was quite low.

*Facebook Activity*. Three quantitative measures of Facebook activity were obtained: The number of Facebook friends (as estimated by the participants), number of hours spent on Facebook per week (Likert scale from (*1*)–(*9*), ranging from *less than one hour (1)* to *more than 20 h (9)*) and log-in frequency [Likert scale from (*1*)–(*9*), ranging from *less than once per month (1)* to *more than five times per day (9)*].

### 2.3. Statistical Methods

IBM SPSS 27 was used to analyze the data. To examine the hypotheses, bootstrapping regression models were employed, which are recommended if assumptions about the normal distribution of scores might be violated or if the number of participants is rather small. Additionally, correlations and partial correlations as well as Fisher Z-tests were calculated to check for significant deviations between correlation coefficients. To assess moderational effects, analyses of variance including the Johnson–Neyman technique were used. Finally, a test of excessive significance (TES) [58] was conducted to check for the median observed power, the success rate, the inflation rate, and the replicability index using the p-checker app (https://shinyapps.org/apps/p-checker/; accessed on 1 March 2021).

## 3. Results

### 3.1. Participants

The participants initially consisted of 118 adults. Most of them were either students graduating at the Ruhr University of Bochum and surrounding universities (e.g., University of Dortmund; University of Hagen). The inclusion criteria were legal age, active Facebook membership, and private (i.e., not only commercial) usage of Facebook. As five participants did not own a Facebook profile, and three participants used Facebook only for commercial purposes, 110 participants were finally included in the statistical analyses. Eighty-two (74.5%) of them used Facebook for private purposes only, whereas 28 (25.5%) used Facebook for both private and commercial purposes. Twenty-three participants (20.9%) were male, 87 (79.1%) were female. Their mean age was 25.52 years (*SD* = 8.149; range: 18–58). All participants were neurotypical German speakers. Most participants stated that their highest educational qualification was A level (either “*Abitur”* or “*Fachabitur”;* 51.8% and 6.4%, respectively); 27.3% had either achieved an academic degree (21.8%) or a successful state examination (5.5%); 10% had completed an apprenticeship; 4% had completed middle school; and one participant (0.9%) had completed secondary school. The majority of participants (65.5%) were currently studying; most of them (44.5%) were students of psychology. Most participants reported logging in on Facebook on at least a daily basis (once per day: 13.6%; twice to thrice: 33.6%; 4 to 5 times: 16.4%; >5 times per day: 12.7%). Only three participants (2.7%) logged in less than once per month. Most of the participants displayed a moderate extent of weekly Facebook usage, i.e., they spent less than four hours per week on the SNS (<1 h: 21.8%; 2 to 3 h: 29.1%; 3 to 4 h: 21.8%). However, nearly one third spent more than five hours on Facebook (5 to 7, and 7 to 10 h: 10%, respectively; 10 to 15 h: 1.8%; 16 to 20 h and >20 h: 2.7%, respectively).

Table 1 summarizes the descriptive statistics of all scales. With one exception, all scales displayed acceptable to good reliabilities, with Cronbach’s α values reaching from 0.71 for the SSE *Performance* subscale to 0.89 for the NI-R. First, with respect to the FAQ the mean scores on the FAQ *Watching* (*M* = 2.33; *SD* = 0.66) and especially *Acting* (*M* = 2.11; *SD* = 0.56) subscales were rather low, in contrast to a rather high mean value on the *Impressing* FAQ subscale (*M* = 3.37; *SD* = 0.88). Second, participants tended to achieve rather low NPI-15 sum scores (*M* = 4.76, *SD* = 3.48). In addition, the Kolmogorov–Smirnov test indicated a significant difference from normally distributed data for the SSE *Appearance* subscale (*p* < 0.05), for which the histogram showed a bimodal distribution. Participants’ NPI-15 and FAQ *Watching* and *Acting* scores displayed a right-skewed distribution. Finally, NI-R and FAQ *Acting* scores displayed a leptokurtic distribution. Because root- as well as square transformations of the skewed data did not improve the shape of the distributions considerably, the level of significance was set at *p* < 0.01 for all hypotheses tests to counteract these limitations. Additionally, bootstrapping was employed for conducting regression analyses because of the resulting robustness of bootstrapping regression models against model violations.

With respect to the priming procedure, all participants completed the ranking order task. The majority (30%) chose a satisfying relationship as their subjectively most important category. Success in achievement domains and social popularity were placed on the first rank by 16.4%, and by 7.3% of respondents, respectively. Participants mostly compared themselves with friends and relatives, followed by acquaintances, colleagues or fellow students, and spouses. Six participants assigned to the upward comparison condition, and one participant assigned to the downward comparison condition did not write down any comparison standards and also did not report their thoughts or feelings during a moment of inferiority or superiority. However, the priming results did not differ significantly as a function of whether the seven non-completers were included or not. Hence, results are subsequently reported including all 110 participants.

Table 2 shows the correlations among the employed scales. GN was significantly and positively associated with SSE (zero-order correlation: *r* = 0.41, *p* < 0.001; controlled for VN: *r* = 0.44, *p* < 0.001), whereas VN was only negatively associated with SSE to a significant extent if GN was controlled for (*r* = −0.25, *p* < 0.01). Among the three SSE subscales, only the *Social* subscale was significantly correlated with GN. The correlation between GN and VN became significant when SSE was partialled out (*r* = 0.26, *p* < 0.01). Furthermore, the FAQ subscales were positively correlated with each other. The *Acting* subscale displayed the highest correlation with the total FAQ.

Supporting previous results, GN and VN were correlated positively [43], indicating the existence of core narcissism which represents both GN and VN. Therefore, the further statistical analyses are based on partial correlations.

In correspondence with the first hypothesis, the results of a partial correlation analysis indicate that grandiose narcissism is positively correlated with SSE (controlling for VN). This pattern of results was confirmed with respect to the NPI for SSE*_Total_*, SSE*_Performance_*, SSE*_Social_*, and SSE*_Appearance_*, constituting strong evidence for H1. In addition, the explained variance was substantial both for the total score and for each of the subscales. Furthermore, the second part of H1 received also some support from the data because the partial correlations of VN with SSE*_Total_* and SSE*_Social_* were significantly negative.

The regression analysis conducted to test whether VN moderates the influence of downward comparison direction on participants’ situational self-esteem (H2) revealed that VN significantly moderated the influence of downward comparison direction on participants’ SSE: The overall fit of the model comprising all eight predictors became significant in predicting SSE *Total* scores (*F* (8, 103) = 5.77, *p* < 0.001), and the same was true for each model with, respectively, one of the three SSE subscale scores as criterion (SSE *Performance*: *F* (8, 101) = 4.35, *p* < 0.001; SSE *Social*: *F* (8, 101) = 3.10, *p* < 0.01; SSE *Appearance*: *F* (8, 101) = 4.01, *p* < 0.001). The entire model explained 31.4% of the variance in participants’ total SSE, and 25.6%, 19.7%, and 24.1% of the variance in the SSE *Performance, Social,* and *Appearance* subscales, respectively. As shown in Table 3, in all of these models, one significant main effect was found: NPI-15 scores significantly, and positively, predicted total SSE (β = 0.08, *p* < 0.001), as well as all SSE subscales (SSE *Performance*: β = 0.08, *p* < 0.01; SSE *Social*: β = 0.09, *p* < 0.05; SSE *Appearance*: β = 0.06, *p* < 0.01). Likewise, in all models—except the one with SSE *Appearance* as criterion, where no significant interaction effect was revealed—one significant interaction effect emerged: namely the downward comparison condition × NI-R interaction: In participants assigned to the downward comparison condition, SSE significantly decreased as NI-R values increased (SSE *Total*: β = −0.65, *p* = 0.001; SSE *Performance*: β = −0.83, *p* < 0.01; SSE *Social*: β = −0.77, *p* < 0.05). Moreover, SSE scores of participants in the upward comparison condition displayed a, though non-significant, trend to decrease in response to an increase in NI-R scores (with significance levels of up to *p* = 0.08, with a corresponding regression weight of β = −0.45, in the model with SSE *Appearance* as criterion). The same was true for the upward comparison condition × NPI-15 interaction effect, although that trend was much less pronounced here and never reached significance levels exceeding *p* = 0.14 (with a corresponding regression weight of β = −0.09, when effect on SSE *Social* was measured). Post-hoc analyses by means of the Johnson-Neyman technique revealed two zones of NI-R scores, within which the conditional effects of downward comparisons on SSE *Total* became significant. One of them included NI-R scores of 1.12 (or 1.71 below mean; β = 0.71, *p* = 0.05, CI [0.01–1.52]) and lower; the other included NI-R scores of 3.2 (or 0.37 above mean; β = −0.26, *p* = 0.05, CI [−0.52–0.00]) and higher. For the SSE *Performance* and *Social* subscales, one region of significance could be determined, respectively, which included NI-R scores of 3.62 (or 0.79 above mean; β = −0.56, *p* = 0.05, CI [−1.12–0.00]) and higher for SSE *Performance,* and of 3.27 (or 0.44 above mean; β = −0.46, *p* = 0.05, CI [−0.91–0.00]) and higher for SSE *Social* as criterion. In all cases, conditioned effects and corresponding significances strongly increased as NI-R scores increased (e.g., up to β = −0.92, *p* = 0.01, CI [−1.64–−0.20], for an NI-R score of 4.59 with SSE*_Total_* as criterion).

Hypothesis 3 referred to the associations of VN with different measures of Facebook activity. As shown in Table 4, VN was significantly correlated with participants’ FAQ-scores. Moreover, the correlations between VN and time spent on Facebook were marginally significant. All other measures of Facebook activity were not significantly correlated with participants’ extent of VN. Zero-order correlations and partial correlations turned out to be quite similar in general.

Hypothesis 4 assumed that GN displays a significant positive association with Facebook activity. An inspection of Table 4 reveals that the assumption was mostly disconfirmed. Only a marginally significant correlation with number of Facebook friends emerged. In general, these results contradict the hypothesis.

Hypothesis 5 postulated that GN would not be linked with Facebook activity after control for VN. The correlations between GN and most measures of Facebook activity were non-significant, and this was true no matter whether VN was controlled for or not (see Table 4). Therefore, H3 was confirmed. We used the Fisher-*Z*-test to compare the correlations between grandiose and vulnerable narcissism and the Facebook Activity Index including number of Facebook log-ins, hours online, and number of Facebook friends. The results are revealing. The overall score of the FAQ (*Z* = 3.729, *p* < 0.001) as well as the FAQ-subscales correlated significantly higher with VN than with GN (FAQ*_Impress_*: *Z* = 4.435, *p* < 0.001; FAQ*_Watch_*: *Z* = 3.430, *p* < 0.001, FAQ*_Act_*: *Z* = 3.123, *p* < 0.001). The correlation difference was marginally significant for log-ins on Facebook (*Z* = 1.373, *p* = 0.085) and significant for number of Facebook friends (*Z* = −2.496, *p* < 0.01) and hours spent on Facebook (*Z* = 1.718, *p* < 0.05). The latter difference indicates that number of Facebook friends was correlated higher with GN than with VN, reversing the general trend.

RQ 1 referred to in how far GN and VN might moderate the influence of upward social comparisons on Facebook activity. First, the ANOVA test for the overall fit of the model to predict participants’ FAQ scores became significant (*F* (8, 101) = 3.96, *p* < 0.001). The same was true for the models employed to predict participants’ FAQ *Watching* and *Impressing* subscales (FAQ *Watching*: *F* (8, 101) = 3.70, *p* = 0.001; FAQ *Impressing*: *F* (8, 101) = 2.17, *p* < 0.05). However, the model with FAQ *Acting* as criterion displayed a non-significant overall fit. Altogether, the eight predictors explained 23.9% of variance in FAQ *Total* scores, and 14.7% and 22.7% of the variance in the *Watching* and *Impressing* subscales, respectively. Only one significant main effect was revealed: NI-R scores significantly predicted FAQ scores (FAQ *Total:* β = 0.45; FAQ *Watching:* β = 0.47; FAQ *Impressing:* β = 0.68; *p* < 0.01, respectively), confirming once again H3. As shown in Table 5, no significant group assignment × narcissism interaction effects were revealed for the FAQ as a whole and the *Impressing* subscale. However, in the model predicting participants’ scores in the *Watching* subscale, a significant negative interaction between NPI-15 scores and exposure to upward comparisons was found (β = −0.12, *p* < 0.01).

This result contains both theoretical and practical importance. Post-hoc analyses revealed that conditioned effects of upward comparisons on FAQ *Watching* became significant for NPI-15 scores of 7.46 (or 2.70 above mean; β = −0.34, *p* = 0.05; CI [−0.69–0.00]) and higher, as visualized in Figure 1. Effects reached up to β = −0.90 (*p* < 0.05; CI [−1.69–−0.11]) for an NPI-15 score of 13.00. None of the other group × VN or group × GN interactions that were tested reached significance with respect to the prediction of FAQ *total* or any FAQ subscale scores.

### 3.2. Replicability

A test of excessive significance based on six hypotheses-oriented effects (i.e., correlational effects, F-statistic, Z-statistic) was conducted. The calculations revealed a success rate of 0.8333, indicating that 83,33% of our hypotheses could be confirmed, and a median observed power of 0.964 was achieved. Above, the TES revealed a deflation (i.e., a negative inflation rate) of −0.1306, indicating that less hypotheses than possible were confirmed with respect to the median observed power. Finally, R-Index = 1.0946, indicating that our findings can be (theoretically) replicated in X*1.0946 follow-up studies. Concluding the results of the TES, the data does not seem to be biased, the power was sufficient despite the fact that the sample was quite small, and the findings seem to be generalizable.

## 4. Discussion

The major aim of the present study was to get deeper insights into how individuals who have been exposed to social comparisons and who display a high extent of narcissism use Facebook to self-regulate. Hypothesis 1 predicted that grandiose narcissism is positively related with state of self-esteem. This hypothesis was confirmed by the result across several measures of state of self-esteem. High grandiose narcissism was systematically associated with high self-esteem. This consistent result replicates the findings of earlier studies [37,42,43]. Grandiose narcissists seem to be on the self-confident side of life. In contrast, VN was negatively correlated with SSE, also confirming previous findings of several studies [42]. Therefore, confirmation of H1 is in correspondence with the idea of two distinct faces of narcissism [39]. Obviously, vulnerability-sensitivity implies low self-confidence, whereas grandiosity-exhibitionism implies high self-confidence.

The second hypothesis, which postulates that VN would moderate the influence of downward comparison direction on participants’ SSE, enters new territory by emphasizing the consequences of perceived own vulnerability on the repercussions of downward comparisons. More specifically, it was predicted that vulnerable narcissists would respond with reduced state self-esteem to downward comparisons. This hypothesis, which was mostly supported by the results of multiple regression analyses, enables a better understanding of the effects of downward comparisons on self-esteem. Because a significant interaction between VN and assignment to the downward comparison condition was discovered in three of the four regression models—namely, in the ones with the entire SSE scale, the SSE *Performance*, and the SSE *Social* subscale as criterion, respectively—the evidence for hypothesis 2 is quite strong: The higher the extent of VN, the lower participants’ SSE in response to having experienced a downward comparison situation. These results correspond with the proposal by Lockwood [29] that participants who compare themselves downward with a target respond with low self-confidence if they think that the bad fate of the target could easily befall them. Imagining failure of others while feeling vulnerable is likely to impair state self-esteem. The focus seems to be on avoiding unfavorable outcomes. The felt vulnerability of vulnerable narcissists which is associated with a lack of self-confidence [39] corresponds with their goal to avoid failure. Because most of the target persons were friends or relatives, in most comparisons the similarity between participant and target was implied.

In correspondence with their defensiveness, the lack of self-confidence and increased anxiety of vulnerable narcissists seem to accentuate the negative implications of poor results by others for themselves. Their slogan seems to be “That could happen to me too!”. That means that vulnerable narcissists seem to suffer negative repercussions on the basis of thinking about the negative fate of others. This is more or less reasonable, but the implications of this sensitivity for own vulnerability are self-threatening [33]. The current results correspond with the existence of a person-situation interaction [59] in terms of VN and downward comparisons. It would be interesting to investigate how other personality traits affect the relationship between exposure to downward social comparisons and state self-esteem. For example, optimism is likely to contribute to an increase in self-confidence on the basis of downward comparisons, whereas pessimism probably will contribute to an impairment of self-confidence in the face of the bad fate of others [60].

In their summary of more than sixty years of research on social comparison processes, Gerber and colleagues [17] come to the conclusion that, in general, downward comparisons elicit positive shifts in self-evaluation, whereas upward comparisons elicit negative shifts in self-evaluation. Beyond this general trend, deviations from the rule occur. Under certain conditions downward comparisons might activate assimilation—and not contrast—with the negative fate of others. In the same vein, upward comparisons are likely to elicit both contrast and assimilation depending on the meaning which the participant attaches to it [24]. For example, upward comparisons might be considered as an incentive to improve own performances or it might be interpreted as inferiority compared with others.

Hypothesis 3 was mostly confirmed as VN displayed a significant positive relationship with measures of Facebook activity. In contradiction with H4, a lack of significant correlations between GN and Facebook activity emerged. The explanation for this result is that grandiose narcissists possess a number of personality traits, intra- and interpersonal strategies and interpersonal skills, which enable them to succeed pretty well in elevating their own self-esteem in the offline world. Hence, it is likely that they do not benefit from SNSs as much as people with difficulties in winning over their audience in the offline world do. However, two specific findings regarding H4 are especially noteworthy. First, the only measure of Facebook activity that was consistently correlated with GN was the number of Facebook friends. This is in correspondence with a number of previous studies [10,11,61,62,63] and can be explained by the nature of grandiose narcissists, who, according to the (extended) agency model [45,46], aim to acquire as many (superficial) friendships as possible and tend to be quite popular in their social environment. Therefore, it is likely that people who know grandiose narcissists in the offline world tend to admire them and become friends with them in the online world. People who know grandiose narcissists offline might simply be keen on being friends with these admirable, charismatic, and much-loved persons. This assumption is also consistent with Brailovskaia and Bierhoff’s [62] finding that, beyond GN, the personality trait of extraversion was highly correlated with users’ numbers of SNS friends.

With respect to H5, which assumed that the association between VN and Facebook activity is stronger than the association between GN and Facebook activity, it is noteworthy that Fisher *Z-*tests revealed that the correlations of NI-R with FAQ are significantly higher than the correlations of NPI with FAQ. Clearly, the level of VN matters when it comes to Facebook use, but the level of GN not so much. The only exception to this rule seems to be number of Facebook friends.

Taken together, the findings obtained in the current study contribute to a confirmation of the SOS-T [13]. Facebook seems to attract vulnerable narcissists due to several features it provides, including comparison with others in a hidden manner, communication in a controlled environment, and, more generally speaking, to self-regulation; this is consistent with former empirical findings [14,15]. Vulnerable narcissists seem to employ features of SNSs for self-regulation. This tendency is in line with their reluctance to use competitive strategies to enhance their self-esteem [40,41,44].

With respect to the research question (RQ 1), the interaction effects of (vulnerable or grandiose) narcissism with upward social comparison were generally weak. But a significant negative effect of GN on FAQ *Watching* scores, subsequent to the exposure to upward comparisons, was recorded. This result indicates that selective avoidance of online social comparisons via Facebook *Watching* by grandiose narcissists in response to the exposure to upward social comparison information occurs. It corresponds with the general trend observed by Gerber and colleagues [17] that upward social comparisons elicit negative consequences of self-evaluation, representing a contrast effect. This negative trend is manifested in the reduction of passive Facebook use only by grandiose narcissists. This result fits into the tendency of grandiose narcissists to feel threatened by the superiority of others, which might impair their heightened self-esteem.

## 5. Limitations

The analyses in the present study were based on an online survey including self-report measures and a priming paradigm. The experimental approach is an advantage of our research design. This research, however, is not without limitations. First, the employment of self-reporting measures constitutes a point of concern, as they might prompt socially desirable responding. Although an experimental design was employed for testing hypothesis 2, most of the tests of the hypotheses on the links between narcissism and Facebook activity rest on a correlational design. Second, whereas our research focused on trait narcissism, it would be interesting to manipulate state narcissism experimentally [64] and to measure the effect of this manipulation on Facebook activity.

Third, the participants do not represent the population of Facebook users in general because psychology students were overrepresented. Therefore, a replication study with more representative Facebook users would be desirable.

Fourth, to include a more sensitive measure of priming effects on Facebook use, the employment of assessments of intentions to use Facebook is recommended. The FAQ and the other Facebook use measures which were employed in this study refer to behavioral reports which might be quite resistant against short-term priming effects. Intentions instead of behavioral reports are more likely to be modified by priming manipulations. Nevertheless, the significant effect of exposure to upward comparisons in interaction with GN which was detected on FAQ *Watching* is meaningful from a theoretical point of view, indicating an avoidance effect of grandiose narcissists after exposure to the superiority of others.

## Figures and Tables

**Figure 1 behavsci-11-00039-f001:**
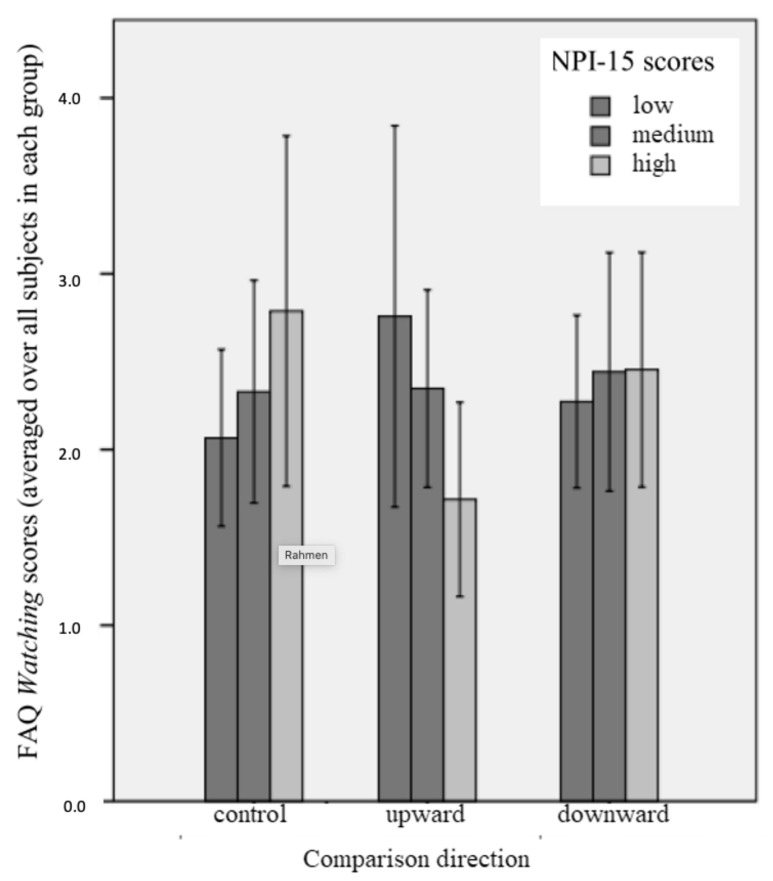
FAQ Watching mean scores depending on group assignment at three levels of NPI-15 scores. Low scores are defined as scores of mean −1 SD and lower. Medium scores are defined as scores above mean −1 SD, but below mean + 1SD. High scores are defined as scores of mean + 1 SD and higher. Error bars represent +/−1 SD.

**Table 1 behavsci-11-00039-t001:** Descriptive statistics of the measures employed in the survey.

Variable	Range	α	*M* (*SD*)	Kur-Tosis	Skew-Ness
Log-ins	>1*/m- <5*/d (1–9)		3.51 (1.92)		
Hours/day	>1/d- <25/d (1–9)		2.90 (1.80)		
Friends	20–3299		346 (374)		
NPI-15	0–15	0.79	4.76 (3.48)	−0.64	0.53
NI-R	1–5	0.89	2.83 (0.51)	2.55	−0.17
FAQ*_Total_*	1–5	0.87	2.44 (0.53)	0.92	0.18
FAQ*_Impress_*	1–5	0.74	3.37 (0.88)	−0.56	−0.33
FAQ*_Watch_*	1–5	0.84	2.33 (0.66)	0.20	0.65
FAQ*_Act_*	1–5	0.75	2.11 (0.56)	1.43	0.47
SSE*_Total_*	1–5	0.87	3.49 (0.63)	−0.36	−0.37
SSE*_Per_*	1–5	0.71	3.82 (0.68)	−0.66	0.10
SSE*_Social_*	1–5	0.82	3.21 (0.88)	−0.56	−0.02
SSE*_App_*	1–5	0.86	3.43 (0.81)	−0.38	−0.22

Note. *M* = mean. *SD* = standard deviation. *N* = 109–110. NPI-15 = Narcissistic Personality Inventory-15 [53]). NI-R = Narcissistic Inventory-Revised [54] FAQ = Facebook Activity Questionnaire [56]. FB*_Impress_*/FB*_Watch/_*FB*_Act_* = *Impressing/Watching/Acting* subscales of the FAQ; SSE = State Self-Esteem Scale [57]. SSE*_Per/_*SSE*_Social/_*SSE*_App_* = *Performance/Social/Appearance* subscales of the SSE.

**Table 2 behavsci-11-00039-t002:** (Partial) correlations among the scales.

Variable	1	2	3	4	5	6	7
1. NPI-15 ^a^	-						
2. NI-R ^a^	0.26 **	-					
3. FAQ*_Total_*	0.09 (−0.01)	0.38 *** (0.37 ***)	-				
4. FAQ*_Impress_*	0.19 * (0.13)	0.19 * (0.25 **)	0.77 ***	-			
5. FAQ*_Watch_*	0.06 (−0.03)	0.34 *** (0.34 ***)	0.79 ***	0.39 ***	-		
6. FAQ*_Act_*	−0.003 (−0.08)	0.27 ** (0.28 **)	0.85 ***	0.56 ***	0.45 ***	-	
7. SSE	0.41 *** (0.44 ***)	−0.16 (−0.25 **)	−0.24 *	−0.21 *	−0.20 *	−0.18	−0.21 *

Note. NPI-15 = Narcissistic Personality Inventory-15 [53]. NI-R = Narcissistic Inventory Revised [54]. FAQ = Facebook Activity Questionnaire [56]. FAQ*_Impress_*/FAQ*_Watch/_*FAQ*_Act_* = *Impressing/Watching/Acting* subscales of the Facebook Activity Questionnaire. SSE = State Self-Esteem Scale [57]. ^a^ The correlations between the narcissism scales and all other scales apart from the SSE are controlled for self-esteem (i.e., the SSE *total* score). The values in brackets constitute the correlation coefficients if the respectively other facet of narcissism is additionally controlled for. * *p* < 0.05; ** *p* < 0.01; *** *p* < 0.001.

**Table 3 behavsci-11-00039-t003:** Linear model of predictors of SSE scales.

Variable	Unstandardized β Coefficients			
	SSE*_Total_* (*R*^2^ = 0.31)	SSE*_Performance_* (*R*^2^ = 0.26)	SSE*_Social_* (*R*^2^ = 0.20)	SSE*_Appearance_* (*R*^2^ = 0.24)
NPI-15 (centred)	0.08 ***	0.08 **	0.09 *	0.06 **
NI-R (centred)	0.06	0.21	−0.19	0.15
SC_Upward_	−0.03	−0.06	−0.15	0.13
SC_Downward_	−0.07	−0.09	−0.22	0.10
NI-R × SC_Downward_	−0.65 **	−0.83 **	−0.77 *	−0.35
NI-R × SC_Upward_	−0.33	−0.49	−0.04	−0.45
NPI-15 × SC_Downward_	0.03	0.04	0.01	0.02
NPI-15 × SC_Upward_	−0.04	−0.04	−0.09	0.00

Note. SC_Downward_, SC_Upward_ = Downward comparison and Upward comparison groups; additionally, “student/no student” was used as a covariate. However, no significant covariation effect occurred (*p* > 0.05). * *p* < 0.05; ** *p* < 0.01; *** *p*< 0.001.

**Table 4 behavsci-11-00039-t004:** Correlations of the two facets of narcissism with measures of Facebook activity.

Variable		*r*, Controlled for			
		Zero-Order	SSE	NPI-15/NI-R	SSE and NPI-15/NI-R
NI-R	FB_Log-ins_	0.18	0.15	0.17	0.13
	FB_Hours_	0.20 *	0.18	0.21 *	0.16
	FB_Friends_	−0.06	−0.06	−0.10	−0.13
	FAQ*_Total_*	0.40 ***	0.38 ***	0.41 ***	0.37 ***
	FAQ*_Impress_*	0.41 **	0.29 **	0.30 ***	0.25 **
	FAQ*_Watch_*	0.36 ***	0.34 ***	0.37 ***	0.34 ***
	FAQ*_Act_*	0.29 **	0.27 **	0.31 ***	0.28 **
NPI-15	FB_Log-ins_	0.02	0.10	−0.01	0.06
	FB_Hours_	−0.00	0.01	−0.04	0.05
	FB_Friends_	0.23 *	0.24 *	0.24 *	0.27 **
	FAQ*_Total_*	−0.02	0.09	−0.10	−0.01
	FAQ*_Impress_*	−0.09	0.19 *	0.04	0.13
	FAQ*_Watch_*	−0.03	0.06	−0.10	−0.03
	FAQ*_Act_*	−0.07	−0.00	−0.13	−0.08

Note. *r* = Pearson correlation coefficient. NI-R = Narcissistic Inventory Revised [54]. NPI-15 = Narcissistic Personality Inventory-15 [53]. FB_Log-ins_ = Log-in frequency. FB_Hours_ = number of weekly hours spent on Facebook. FB_Friends_ = number of Facebook friends (recoded into nine categories). FAQ = Facebook Activity Questionnaire [56]. FAQ*_Impress_*, FAQ*_Watch_*, FAQ*_Act_*, FAQ*_Total_* = *Impressing*, *Watching*, and *Acting* subscales of the FAQ. * *p* < 0.05; ** *p* < 0.01; *** *p* < 0.001.

**Table 5 behavsci-11-00039-t005:** Linear model of predictors of FAQ scales.

Variable	Unstandardized β Coefficients			
	FAQ*_Total_* (*R*^2^ = 0.24)	FAQ*_Watching_* (*R*^2^ = 0.27)	FAQ*_Acting_* ^1^ (*R*^2^ = 0.14)	FAQ*_Impressing_* (*R*^2^ = 0.15)
NPI-15 (centred)	0.01	0.04	−0.01	0.03
NI-R (centred)	0.45 **	0.47 **	0.32 *	0.68 **
SC_Upward_	0.02	−0.34	0.68	0.03
SC_Downward_	0.06	0.00	0.67	0.15
NI-R × SC_Downward_	0.13	0.17	0.19	−0.06
NI-R × SC_Upward_	−0.33	−0.19	−0.27	−0.73
NPI-15 × SC_Downward_	−0.03	−0.07	−0.00	−0.01
NPI-15 × SC_Upward_	−0.07	−0.12 **	−0.02	−0.06

^1^ The model with FAQ *Acting* as predictor displayed a non-significant overall fit. Note. SC_Downward_, SC_Upward_ = Downward comparison and Upward comparison groups; “student/no student” as a covariate was added. However, no significant covariation effect occurred (*p* > 0.05). * *p* < 0.05; ** *p* < 0.01.

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
