# Peer review of "How Downward and Upward Comparisons on Facebook Influence Grandiose and Vulnerable Narcissists’ Self-Esteem—A Priming Study"

_behavsci, 2021, doi:10.3390/bs11030039_

Round 1
Reviewer 1 Report
The paper has got a very suggestive title and a promissing idea about a very actual topic as personality traits & use of social media. I have some questions, comments and suggestions for the authors:
Abstract: there is no mention to the methods in the abstract. It would be proper to include one sentence describing briefly the methodology of the study
Introduction: line 23 is reiterative ("This study investigates social network site (SNS) usage by investigating narcissism").
In the Introduction (but not only in it) there is a clear abuse of self citations. I think it´s better not to justify only with the authors´previous studies the reasons of the study.
In line 60 (but not only in it) the reasons of the study are indicated in first person (We). Suggestion to use third person in redaction.
The description of the hypotheses and the measures (lines 144-214) can be better included in the Methods part of the paper.
Methods: There is a clear limitation in the size of the sample, it´s very small to develop general conclussions about personality traits and use of social media. There is a clear bias in the selection of the sample, most of them are students. Have you done any kind of analysis to decrease/control this bias?
Data from the sample are not part of Results? (lines 241-258)
There is a lack of information in line 262 ("Participants were recruited via flyers distributed at XXX via Facebook posts and personal 262 addresses"). I totally understand the data protection and confidenciality about the study, but if you have only distributed the flyers in one city/town/village/university, it´s necessary more information about how had you controlled from internal validity to external validity of the study.
In Methods part there is a lack of information. What analysis did you make? What statystics did you use? What tool did you use?
Results:
Lines 377-379 are part of the discussion, connecting results with hypotheses.
Data in text are same than data in table, so it´would be more feasible for the reader to find in the text only significative data.
Lines 499-509 are part of the discussion, connecting results with hypotheses.
Discusion.
To support your discussion, it´s necessary to make reference to more recient studies (lines 588, 592).
In the Discussion part, again, abuse of self-citation. The authors can´t use only their own research to justify parts of your study (lines 655, 656, 658)
Author Response
Dear reviewer,
many thanks for your time and effort by reviewing our manuscript. It profited a lot by your work!
Reviewer comments and authors responses:
The paper has got a very suggestive title and a promissing idea about a very actual topic as personality traits & use of social media. I have some questions, comments and suggestions for the authors:
Abstract: there is no mention to the methods in the abstract. It would be proper to include one sentence describing briefly the methodology of the study
Revision:
Many thanks! We revised the abstract and added the methods.
Introduction: line 23 is reiterative ("This study investigates social network site (SNS) usage by investigating narcissism").
Revision:
We changed the wording.
In the Introduction (but not only in it) there is a clear abuse of self citations. I think it´s better not to justify only with the authors´previous studies the reasons of the study.
Revision:
We added additional references (in the intro as well as in the discussion section).
In line 60 (but not only in it) the reasons of the study are indicated in first person (We). Suggestion to use third person in redaction.
Revision:
We reformulated those phrases in third person.
The description of the hypotheses and the measures (lines 144-214) can be better included in the Methods part of the paper.
Revision:
We included this section in the methods.
Methods: There is a clear limitation in the size of the sample, it´s very small to develop general conclusions about personality traits and use of social media.
Revision:
Many thanks you are right! Our sample was quite small. To counteract this limitation, we calculated the test of excessive significance (TES; Schimmack, 2016) to check for the median observed power, the success rate, the inflation rate, and the replicability index using the p-checker app (https://shinyapps.org/apps/p-checker/). The calculations revealed a success rate of .8333 indicating that 83,33% of our hypotheses could be confirmed and a median observed power of .964 was achieved. Above, the TES revealed a deflation (i.e., a neg-ative inflation rate) of -.1306 indicating that less hypotheses than possible were confirmed with respect to the median observed power. Finally, the r-index was R-Index = 1.0946 indicating that our findings can be (theoretically) replicated in X*1.0946 follow-up studies. Concluding the results of the TES, our data does not seem to be biased, the power was sufficient despite the fact the sample was small, and, our findings seem to be generalizable.
There is a clear bias in the selection of the sample, most of them are students. Have you done any kind of analysis to decrease/control this bias?
Revision:
We conducted covariation analysis of studentical status in the main analysis. However, no significant effect occurred.
Data from the sample are not part of Results? (lines 241-258)
Revision:
We changed the sample statistics to the results section.
There is a lack of information in line 262 ("Participants were recruited via flyers distributed at XXXvia Facebook posts and personal 262 addresses"). I totally understand the data protection and confidenciality about the study, but if you have only distributed the flyers in one city/town/village/university, it´s necessary more information about how had you controlled from internal validity to external validity of the study.
Revision:
We now added the University. Before we used the XXX because of the blind peer review process. The flyers were distributed to the Ruhr University of Bochum as well as to neighbor universities (Dortmund, Hagen, Cologne).
In Methods part there is a lack of information. What analysis did you make? What statystics did you use? What tool did you use?
Revision:
We added a section to statistical methods.
Results:
Lines 377-379 are part of the discussion, connecting results with hypotheses.
Revision:
We changed this phrase to the discussion.
Data in text are same than data in table, so it´would be more feasible for the reader to find in the text only significative data.
Revision:
We revised this section.
Lines 499-509 are part of the discussion, connecting results with hypotheses.
Revision:
We changed this phrase to the discussion.
Discussion.
To support your discussion, it´s necessary to make reference to more recient studies (lines 588, 592).
Revision:
We tried to add more recent studies. Thank you.
In the Discussion part, again, abuse of self-citation. The authors can´t use only their own research to justify parts of your study (lines 655, 656, 658)
Revision:
We added additional references.
Reviewer 2 Report
Article 117235
“If you wonder why into Facebook I delve, the answer's my vulnerable self!- About Facebook, narcissism, and social comparisons”
The present study aimed to investigate how grandiose narcissism and vulnerable narcissism as well as social comparisons are associated with Facebook activity, which was measured on three dimensions: Acting, Impressing, and Watching.
I first want to thank the authors for their interesting contribution.
The Authors can change the title with another form that reflect more the complexity of the study and is aim.
The Authors should revise the format of the text (for example lines 573-576, lines 612-618, lines 667-671, lines 844-845)
Line 237
The Authors have to change the term “sample” with “participants”, in line with recent recommendation form APA.
Results
The Authors should try to simplify the tables, maybe too complex. I suggest do in a readable way.
References
The authors should revise the references in the text and ate the end of the article (for example Błachnio, A., Przepiórka, A., & Rudnicka, P. (2013). Psychological determinants of 704 using Facebook: A research review. International Journal of Human-Computer Interac-705 tion, 29(11), 775-787. 706;
Błachnio, A., Przepiorka, A., & Rudnicka, P. (2016). Narcissism and self-esteem as 707 predictors of dimensions of Facebook use. Personality and Individual Differences, 90, 296-708 301.in the text is written in another way)
Author Response
The Authors can change the title with another form that reflect more the complexity of the study and is aim.
Revision:
We reformulated the title.
The Authors should revise the format of the text (for example lines 573-576, lines 612-618, lines 667-671, lines 844-845)
Revision:
We revised the format. Thanks!
Line 237
The Authors have to change the term “sample” with “participants”, in line with recent recommendation form APA.
Revision:
Revised.
Results
The Authors should try to simplify the tables, maybe too complex. I suggest do in a readable way.
Revision:
We tried to simplify our tables. Thank you a lot!
References
The authors should revise the references in the text and ate the end of the article (for example Błachnio, A., Przepiórka, A., & Rudnicka, P. (2013). Psychological determinants of 704 using Facebook: A research review. International Journal of Human-Computer Interac-705 tion, 29(11), 775-787. 706;
Błachnio, A., Przepiorka, A., & Rudnicka, P. (2016). Narcissism and self-esteem as 707 predictors of dimensions of Facebook use. Personality and Individual Differences, 90, 296-708 301.in the text is written in another way)
Revision:
The references were revised.
Dear reviewer,
many thanks for your time and effort spent in this manuscript. It helped us to revised our paper by a lot.
Reviewer comments and authors responses:
Reviewer 3 Report
Dear Authors,
Please, see the attached file.

Author Response
Major Issues
TITLE
- The current title is catchy but not very comprehensive. An alternative and more systematic title focused on the primary objective of the research are suggested.
Revision:
We formulated a new title.
ABSTRACT
- The abstract mainly contains the aim and results. It can be enhanced as follows:
- Write one starting sentence which introduces the reader to the association between
Facebook and narcissism.
- A sentence on the hypothesis should appear after the sentence, which explains the aim of
the research.
- The abstract should also clarify how Facebook activity was measured.
- Make direct which result was in line with the hypothesis. Avoid explaining the hypothesis
when reporting the results, as in the current version.
- Authors should maintain VN and GN's abbreviation throughout all the abstract if they decide
to adopt them.
- If possible, write a final sentence which summarizes the conclusions of the research.
Revision:
We reformulated the abstract.
INTRODUCTION
The first sentence of the introduction (lines 23-26) should very briefly and generally
announce the constructs involved in the research (with no direct reference to the
investigation). In the current version, the first sentence seems unproperly to include the first
aim of the research. The aim is already well-described at the end of the introduction.
Revision:
We reformulated this section.
- Lines 60-63. The distinction between upward and downward comparison is not yet
explained. Upward and downward comparison is here mentioned for the first time. What is
the difference between the two conditions? Why are they adopted for the experimental
manipulation? A line that connects Facebook activity, self-regulation, and self-esteem
through the upward and downward comparisons should clarify this point.
Revision:
We added these information. Many thanks for your recommendation.
Alternatively, lines 60-63 could advance more general consideration on Facebook activity, self-regulation and self-esteem since upward and downward comparison are not yet defined until lines 74-77.
Revision:
We revised this section. Thanks.
- Lines 74-77. Please, help the reader to understand what downward and upward comparison
stand for. How do people compare downward and upward with others? How downward and
upward comparison apply to Facebook users? This point becomes evident only from the
methods and the discussion.
Revision:
We added some paragraphs to apply to this point.
- Lines 149-181. The preliminary introduction to hypothesis 2 is significantly extended. A
briefer and more careful consideration, able to guide the reader into the hypothesis is
suggested.
Revision:
We shorted this section.
- Lines 186-205. The preliminary introduction to hypotheses 3 is quite long and includes some
of the content already presented. A succinct and comprehensive explanation should be
required to connect hypothesis 2 with hypothesis 3.
Revision:
We shorted this section.
METHODS
- A new paragraph which explains the performed descriptive and inferential statistical analysis
(Pearson's r, regression models, Fisher's z, Johnson-Neyman, ANOVA) is highly
recommended. Specify also the software and its version.
Revision:
We added a section with respect to statistical methods.
- The authors do not justify how they set the sample. A post-hoc statistical power would be
required for each primary statistical test.
Revision:
Many thanks you are right! Our sample was quite small. To counteract this limitation, we calculated the test of excessive significance (TES; Schimmack, 2016) to check for the median observed power, the success rate, the inflation rate, and the replicability index using the p-checker app (https://shinyapps.org/apps/p-checker/). The calculations revealed a success rate of .8333 indicating that 83,33% of our hypotheses could be confirmed and a median observed power of .964 was achieved. Above, the TES revealed a deflation (i.e., a neg-ative inflation rate) of -.1306 indicating that less hypotheses than possible were confirmed with respect to the median observed power. Finally, the r-index was R-Index = 1.0946 indicating that our findings can be (theoretically) replicated in X*1.0946 follow-up studies. Concluding the results of the TES, our data does not seem to be biased, the power was sufficient despite the fact the sample was small, and, our findings seem to be generalizable.
- It is not clear how authors managed the "multiple comparison problem" and adopt one of
several methods used to control statistical inferences by repeated tests (e.g., correlations,
linear models).
Revision:
To counteract this, we now calculated the replicability analyses as well as power analyses and, for this, we set the p-limit to .01.
RESULTS
- Lines 354-359. Why was the level of significance adjusted at .01 a-priori? Authors could
consider improving the skewed distributions by square-transformation, a highly used
method for subscales that do not show a Gaussian curve.
Revision:
Many thanks for your recommendation. We tried to resolve the problem with root as well as root-square transformations. However, because root- as well as square transformations of the skewed data did not improve the shape of the distributions considerably, the level of significance was set at p < .01 for all hypotheses tests to counteract these limitations. Additionally, bootstrapping was employed for conducting regression analyses because of the resulting robustness of bootstrapping regression models against model violations.
DISCUSSION
- Lines 637-641. The explanation offered here contrasts with the antecedent justification of
the results (lines 630-637). Authors probably meant that people who know grandiose
narcissists in the offline world tend to admire and become friends in the online world. Please,
clarify.
Revision:
We reformulated this section.
Minor Issues
- Lines 29-34. These sentences are not useful for this work, which is not exploring Facebook
development from its origins. Authors could replace with a unique and succinct sentence.
Revision:
Revised.
- Line 56. Unravel the concept of materialism applied to Facebook users.
Revision:
We added a sentence unraveling the concept of materialism.
- Line 71. Authors could explain better what they mean by "…the choice of standards…".
Revision:
We added an explanation.
- Line 108. Authors could clarify what they mean by "…in a category that is subjectively
important to the participants".
Revision:
We added an explanation.
- Line 111. The paper should be self-explanatory. Please, remove the final part of the sentence
(e.g., "…as further elaborated in the following section")
Revision:
We removed this formulation.
- Lines 146-148. Authors could explain better what they mean by "…after statistically
controlling for" the other condition (vulnerable or grandiose).
Revision:
We added an explanation.
- Line 156. Revise "keep in mind".
Revision:
revised.
- Lines 160-162: very long sentence. Please, rephrase.
Revision:
We rephrased the sentence.
- Add a link between line 158 and the line 159. It is not clear how authors move from GN's
considerations, VN and downward comparison to the consequences of vulnerability on selfratings
after downward comparisons.
Revision:
We added a link.
- Line 189-190. The following sentence is not exact: "We focused on situational self-esteem
which should be more malleable by situational manipulations than general self-esteem."
What do the authors mean? Please, explain.
Revision:
We tried to explain it more precisely.
- Lines 207-214. Authors could explain better what they mean practically with the following
expressions: "… control for the common core of narcissism…" or again "… partialling out the
common core of the narcissism."
Revision:
We tried to explain it more precisely.
- Line 239. Please report the name of the "University XXX".
Revision:
We now added the University. Before we used the XXX because of the blind peer review process. The flyers were distributed to the Ruhr University of Bochum as well as to neighbor universities (Dortmund, Hagen, Cologne).
- Were all the participants neurotypical native German?
Revision:
We added a sentence. They were all neurotypically German.
- Please, write some items of example for the English NPI and NI.
Revision:
We added some verbatim items.
- Please, adjust the second column of Table 1 to facilitate reading.
Revision:
We tried to simplify the table.
- Lines 596-598. The following sentence is not immediately clear: "the case of vulnerable narcissists is different because they seem to accentuate the negative implications of poor results by others for themselves." Please, rephrase.
Revision:
We rephrased it.
- Lines 630-632. Part of the sentence is not self-explanatory: "…regardless of whether VN was partialled out or not,…". Please, explain.
Revision:
We tried to explain it more precisely. Thank you a lot!
Round 2
Reviewer 1 Report
I would like to start congratulating the authors because this second version of the study has changed a lot of core aspects of the study, such as the title, abstract, background and methods. I have only two comments about the paper:
- There is repeated information in the paper, such as Hypotheses (lines 166-234 and lines 254-309). My suggestion is to include this information only once, or at the end of Background or at the beginning of Methods.
- There is, in my opinion, a clear abuse of self-citation in the text. It´s true the effort of authors to integrate the self-references into a bigger group of bibliography, but there are still parts in the paper sustained by previous studies of the authors (Background and Discusion, mainly).
There are formal aspects such as the change of format in lines 330-345 (sample) and the use of first person ("we...") in some moments of the text (lines 350, 441, 444, 658, 700, 719, 839). I recommend to make these small changes for the publication of the paper.
Author Response
- There is repeated information in the paper, such as Hypotheses (lines 166-234 and lines 254-309). My suggestion is to include this information only once, or at the end of Background or at the beginning of Methods.
--> many thanks. We rearranged the hypotheses section by shortening and putting some paragraphs in the intro.
- There is, in my opinion, a clear abuse of self-citation in the text. It´s true the effort of authors to integrate the self-references into a bigger group of bibliography, but there are still parts in the paper sustained by previous studies of the authors (Background and Discusion, mainly).
--> We again tried to resolve this problem and hope you will agree with our revisions.
There are formal aspects such as the change of format in lines 330-345 (sample) and the use of first person ("we...") in some moments of the text (lines 350, 441, 444, 658, 700, 719, 839). I recommend to make these small changes for the publication of the paper.
--> Thank you! We changed everything from first to this person! We also revised the formatting problem so far.
Reviewer 3 Report
Dear Authors,
Thanks for your work. The paper has been considerably improved.
Please, consider the following point.
I observed that you included the hypothesis in a new paragraph of the "Methods", as required by another reviewer.
However, most of the content seems an exact repetition of what already reported at the end of the "Introduction" (e.g., see lines 272-278, 292-296, 310-322). It seems that it does not include additional details concerning the methodological approaches.
To solve this issue, possible options could be:
- Explain the hypotheses only in the "Methods" and include only the theoretical framework in the "Introduction". If this is the choice, authors could also link the content with the hypotheses, reporting the number of the hypothesis to which the hypothesis is referred (for instance, line 224 could be "The present study aimed to confirm and extend this result with the formulation of a hypothesis across three predictions (see HP3a, HP3b, HP3b in the section Hytpotheses)"
- Alternatively, keep the hypotheses both in the "Introduction" and in the "Methods", but avoiding repetitions and differentiating the two paragraphs.
Author Response
However, most of the content seems an exact repetition of what already reported at the end of the "Introduction" (e.g., see lines 272-278, 292-296, 310-322). It seems that it does not include additional details concerning the methodological approaches.
--> Thank you so much. We shortened the hypotheses section and arranged this section new. The ms. profited a lot by your recommendations!